# The Effect of Height on Drop Jumps in Relation to Somatic Parameters and Landing Kinetics

**DOI:** 10.3390/ijerph17165886

**Published:** 2020-08-13

**Authors:** Krzysztof Mackala, Samo Rauter, Jozef Simenko, Robi Kreft, Jacek Stodolka, Jozef Krizaj, Milan Coh, Janez Vodicar

**Affiliations:** 1Department of Track and Field, University School of Physical Education, Ul. Paderewskiego 35, 51-612 Wrocław, Poland; jacek.stodolka@awf.wroc.pl; 2Faculty of Sport, University of Ljubljana, Gortanova ul. 22, 1000 Ljubljana, Slovenia; Samo.Rauter@fsp.uni-lj.si (S.R.); robi.kreft@fsp.uni-lj.si (R.K.); jozef.krizaj@fsp.uni-lj.si (J.K.); Milan.Coh@fsp.uni-lj.si (M.C.); Janez.Vodicar@fsp.uni-lj.si (J.V.); 3Essex Pathways Department, University of Essex, Wivenhoe Park, Colchester CO4 3SQ, UK; j.simenko@essex.ac.uk

**Keywords:** counter-movement, kinetics, explosive power, force production

## Abstract

The aim of this study was to assess the effect of drop height and selected somatic parameters on the landing kinetics of rebound jumps in force and power production, performed by male and female student athletes. Twenty female and forty male students with a sports background participated in the experiment (mean and standard deviation (± SD): age 20.28 ± 1.31 years, height 166.78 ± 5.29 cm, mass 62.23 ± 7.21 kg and 21.18 ± 1.29, 182.18 ± 6.43, 78.65 ± 7.09). Each participant performed three maximal jumps on two independent and synchronized force platforms (Bilateral Tensiometric Platform S2P) at each of the two assigned drop-jump heights (20-, and 40-, cm for female and 30-, and 60-, cm for the male special platform). Significant between-sex differences were observed in all variables of selected somatics, with men outperforming women. Statistically significant differences were noted in four parameters, between men and women, in both DJs from 20/40 and 30/60 cm. The height of the jump was 6 cm and 4 cm higher for men. A slightly higher statistical significance (p = 0.011) was demonstrated by the relative strength (% BW) generated by the left limb in both men and women. Only women showed a significant relationship between body mass, body height, and five parameters, dropping off of a 20 cm box. In men, only the left leg—relative maximal F (p =−0.45)—showed a relationship with body mass. There were no relationships between the above-mentioned dependencies in both groups, in jumps from a higher height: 40 cm and 60 cm. From a practical application, the DJ with lower 20/30 cm or higher 40/60 cm (women/men) respectively emphasizes either the force or power output via an increase in the velocity component of the rebound action or increased height of the DJ jump.

## 1. Introduction

In most sports, the physical ability of an athlete is to produce explosive power in any dynamic movement or activity in a quick and forceful manner [1,2,3,4,5]. These forceful movements guarantee optimal performance but in some cases, maximal performance in his/her sport. The most powerful movements such as the initiation of movement (start), short acceleration, quick deceleration, cutting maneuverer, and quick change of direction, as well as any type of jumping, require force production into the ground for any propulsion (forward, backward, side to side) [6]. In addition, when an athlete executes any type of jumping activities, they need to limit the impact of force during the landing phase. Therefore, nearly every jump, which involves the application of a counter-movement phase requires the production and absorption of force [5,7,8].

The assessment of explosive power qualities in an athlete will provide objective information for coaches and professionals of their capability to train and perform optimally [9,10,11] and most importantly, to reduce injury risk [12,13]. The assessment concerns not only elite athletes, but all who train regularly, and where ballistic movements occur. Therefore, jumping tasks with counter-movement which rely on the ability to achieve high levels of force via engagement of the stretch-shortening cycle (SSC) will play an important role as tests [14,15,16]. In other words, the tests are a consequence of being an ecologically valid method of assessment if an athlete performs jumping actions in their sport. ‘That is why vertical jumping tests are of great importance in sport: to monitor training progress, to test the level of force development and foremost, to assess the possibilities of maximum performance. Additional benefits include them being time-efficient, easy to administer, and they can be performed with a wide range of equipment; thus, are accessible to all practitioners. There are two distinctly different vertical-jump tests that can be used to assess explosive power performance capacities [17]. The classical counter-movement jump (CMJ) with application of upper arm movement and the drop-jump test, where the counter-movement phase appears in the first part of the exercise (landing) and in the next phase, take-off, to perform a vertical displacement (jump up) of the athlete’s body.

The action where athletes need a quick transition from eccentric to concentric actions with effective use of the SSC is an essential motor skill in many sports [18]. It is expressed through the jump, where jump height or time of execution are decisive for success or failure of a sporting action [19]. According to Pietraszewski and Rutkowska-Kucharska [20], the range of the muscle stretch depends on the height of the box from which the squat jump is performed. The time from landing after the jump to the take-off phase describes the duration of the stretch–shortening cycle. In fact, these two elements really define the type of jump. In the literature, you can find several terms for this type of jump: drop jump, depth jump, bounce jump or shock jump. Despite the popularity of these jumps, the effect on the development of the counter-movement jump height is often inconsistent; however, the jump height is considered the main performance output [21]. The DJ is also debatable as some studies using the test output the Reactive Strength Index (RSI), which is a measure of reactive jump capacity and displays how an athlete copes with and performs plyometric activities [22]. This experiment was based on the so-called drop jump, where the emphasis is placed on a short ground-contact time, less than 0.25 s [17,22,23,24] and with low magnitudes of leg flexion [25]. In a DJ, resistance is added to the counter-movement phase by stepping off, falling, and landing on the ground from a box or platform between 20 and 60 cm [17,25]. Additionally, the target outcome of this exercise is the development of fast SSC from the muscle-tendon units of the lower extremity extensors.

Any differences observed in jump characteristics regardless of the type, horizontal or vertical between female and male and female athletes, may be of interest in any sport. Quantifying any variation, especially anthropometric and possibilities of lower extremity power generation between the sexes may impact training procedure for motor development, injury prevention, and be a tool for talent identification, especially in sports where powerful movements are a major requirement. Recently, some studies have compared the differences in jump performance between the two groups [6,8,26,27,28,29,30,31]. Usually, the differences concerned vertical performance. Abian et al. [32] found a difference of 10 cm between men and women during a vertical jump. The difference in performance could be explained by the difference between the force parameters [33]. Vertical jump performance depends on the vertical velocity at the take-off, which is correlated with the power output [34]. When a great force is required to be applied to the ground during the jump, men are more likely than women to generate this force [32,35,36]. However, according to Komi and Bosco [37], women seem to have better use of the transfer of energy. They use a larger percentage of the energy stored during the pre-stretching phase of jumps than men. 

In turn, not much research has focused exclusively on DJ using either athletes or physical education students [38,39]. It was hypothesized that homogenous groups, either female or male, might progress similar side-to-side (right-to-left) differences in force production that may occur during double-leg rebound jumps performed from diverse drop height boxes. Therefore, the aim of this study was to assess the effect of drop height and selected somatic parameters on landing kinetics of rebound jumps with particular emphasis on side-to-side leg differences in force and power production, displacement-time, and other variables performed by male and female athletes.

## 2. Materials and Methods

### 2.1. Participants

Twenty female and forty-four male physical education students who were strength and conditioning majors participated in the experiment (mean ± SD: age 20.28 ± 1.31 y, height 166.78 ± 5.29 cm, mass 62.23 ± 7.21 kg and age 21.18 ± 1.29, height 182.18 ± 6.43, mass 78.65 ± 7.09). All recruited students were drawn from a variety of sporting backgrounds (university and local sports clubs). Participants took part in regular training 4–5 times per week in their sports discipline (track and field, soccer, tennis, handball, judo, basketball, volleyball) performing strength and some plyometric activities, in accordance with the requirements of their sport. All participants were free from any lower extremity injuries that could affect their jumping performance at the time of testing. This experiment was approved by the review board. The study design was approved by the Human Ethics Committee of the University of Ljubljana (Code:14_2019-1436). The procedures were in accordance with the Code of Ethics of the World Medical Association (Helsinki declaration of 1964). Before signing informed consent forms, the participants were informed about the goal of the experiment and the risk of injury.

### 2.2. Design 

A repeated measure design of two different drop jumps (DJ) was used to investigate the kinetic variables of unilateral vs. bilateral feet, maximal force on landing, and jumping performance between female and male student athletes. Participants were assigned to two groups (men or women) where either unilateral or bilateral landing during jump performance was executed and analyzed. All subjects were familiarized with the testing protocols before the testing session. Athletes were required to take part in one familiarization and one preliminary measurement session. During this measurement, participants performed two DJs from a 20/40 cm platform for females and 30/60 cm for males. The choice of the drop height was not accidental. In order to maximize SSC stimulus, the drop height cannot be too low or too high [40]. In order to increase jump height via increased hip and knee extension, the recommended drop height should be above 20 cm but not exceeding 60 cm [41,42] due to increased injury risk [43]. Considering these facts and the fact that there is an overall difference in lower limb strength between men and women, it was decided that the maximum jump height for women would be 40 cm and for men, 60 cm. During the familiarization session, athletes were instructed in the specific technical requirements of the SJ, CMJ, and DJ. They were allowed enough attempts to make sure they were able to do the jumps technically and safety. All testing was conducted in the afternoon (4–7:00 PM) on a Monday, after a 48 h break from any physical activities. In the first part of the experiment, the participants performed a structured dynamic warm-up, finished with 2–3 DJ from a 20/40 cm drop height. After a 15 min break, they continued and performed two drop jumps. All measurements were performed on both legs simultaneously and each leg separately.

### 2.3. Measurement Procedures

#### 2.3.1. Somatic Measurements

Body height was measured with an anthropometer GPM (DKSH Switzerland, Ltd, Zurich, Switzerland). Measurements of body composition were performed using bioelectrical impedance analysis (BIA), with the InBody 720 Tetrapolar 8-Point Tactile Electrode System (Biospace Co., Ltd. Seoul, South Korea). The InBody 720 apparatus utilizes technology for measuring body composition by using the method of Direct Segmental Multi-Frequency Bioelectrical Impedance Analysis. With InBody 720, we measured body weight, body mass index (BMI), skeletal muscle mass (SMM), and body fat mass (BFM).

#### 2.3.2. Drop-Jump Kinetic Measurements

Each participant performed three maximal jumps on two independent and synchronized force platforms (Bilateral Tensiometric Platform S2P, Ljubljana, Slovenia) at each of the two assigned drop-jump heights (20–, and 40-, cm for females and 30-, and 60-, cm for the male special platform). Jumps were separated by 1 min of rest and 15 min of recovery between drop-jump heights. To initiate the drop movement, participants were instructed to “step out“ from the edge of the platform. In order to avoid jumping from the platform during the dropping movement, they held upper limbs in a controlled position along the body. The landing was simultaneously on both feet, left on one platform, and right on the other. After landing, the participants performed a classic CMJ with dynamic work of the upper limbs. The participants were instructed to jump as “fast as possible”—perform take-off from the ground (tensiometric platform) in the shortest period of time, and as high as possible. For the trial to be valid, participants had to remain on the mat after landing. The jump was invalid if the participant raised feet (tuck) during the CMJ flight or landed behind the platform. The Bilateral Tensiometric Platform S2P was interfaced with computerized software. A dynamic measuring system ARS (Analysis & Reporting Software; S2P Ltd., Ljubljana, Slovenia) was used to collect all drop-jump data. Measurements of flight time and contact time were recorded in milliseconds.

### 2.4. Statistical Analyses

Data analysis was performed with SPSS for Windows 15.0 (IBM, Armonk, NY, USA). Descriptive statistics (mean ± SD) were calculated for all dependent variables. Preliminary analyses were performed for all data to ensure that requirements for parametric testing were met. To test for differences in dependent variables, linear analysis of variance (ANOVA) models were applied. Each ANOVA model was composed of a two-way analysis to test for within-subject differences across the independent variable (i.e., drop height). Within-subject differences were treated as repeated measures. Assumptions of the test statistic were verified. The test–retest reliability of all dependent variables for each drop height was evaluated by intraclass correlation coefficients (ICC). All seven variables were found to have strong reliability with the highest values obtained for jump height (r = 0.93) and contact time (r = 0.91. In turn, relative maximal F and relative maximal P showed the strongest reliability (r = 0.85 and r =0.88 respectively). ICC for relative E were r = 0.86. Effect sizes were evaluated by calculating Cohen’s d with 95% confidence intervals. Cohen suggested that d = 0.2 be considered a ‘small’ effect size, 0.5 represents a ‘medium’ effect size, and 0.8 a ‘large’ effect size. Pearson’s product–moment correlation analysis was used to assess the linear relationship between the somatic and kinematic variables. The criterion for statistical significance was set at an alpha-level of 0.05.

## 3. Results 

Significant between-sex differences were observed in all variables of selected somatic variables with men outperforming women (Table 1). The greatest absolute difference between male and female subjects was observed in body height, body mass, SMM, and percent body mass. 

Table 2 presents a comparison of selected CMJ kinetic parameters as a result of jumps from two different heights, between men and women. Women jumped from 20 cm and men from 30 cm. Statistically significant differences were noted in four parameters, one of which concerned the height of the jump, which was 6 cm higher for men, and the other parameters determined the differences in the force generated on the ground. A significant difference (p = 0.00) occurred in the level of relative strength obtained from the rebound, which was 9.66% higher in men compared to women. A similar relationship was observed when the same jump was divided into the measurement of relative force, taking into account the left and right limb. A slightly higher statistical significance (p = 0.011) was demonstrated by the relative force (% BW) generated by the left limb.

In turn, Table 3 presents a comparison of selected dynamic parameters of CMJ—rebound of the same groups, but taking into account the higher fall height—men 60 cm, women 40 cm. The higher drop-off caused significant differences (in favor of men) also in four parameters (the same as in Table 1). The time of foot contact with the ground during landing/rebound was at the border of significance and was p = 0.069. The other two parameters, relative maximal P [W/kg] and relative E [J/kg], showed no significant differences. The trend from the previous analysis (Table 1)—the difference between the relative strength in the ratio of the left to the right limb, men to women—was preserved. 

Five kinematic parameters showed a significant relationship (Table 4) with body mass and four parameters with body height in women dropping-off of a 20 cm box. In men only the left leg relative maximal F (p = −0.45) showed a relationship with body mass. There were no relationships in both groups for jumps from a higher height: 40 cm and 60 cm, for women and men, respectively.

## 4. Discussion 

The DJ with different heights of drop-off and with counter-movement phase (rebound) is a commonly used tool to explore differences in the neuromuscular function of plyometrics. Previous studies have quite extensively reported gross CMJ measures; however, there are not many explored sex differences in these characteristics. Therefore, the aim of this study was to assess the effect of drop height and selected somatic parameters on the landing kinetics of rebound jumps in force and power production, performed by male and female athletes.

Despite the popularity of any kind of DJ type exercise [44] as a tool of lower extremity maximal power output development, men jumped ~21% and ~14% higher than women in jumps from both 30 and 60 cm drop-offs compared to the 20/40 cm box, comparable with other research done by Laffaye et al. [8] and Rice et al. [31]; however, athletes achieved much better results. Cited authors [8.31], claimed that the difference between men and women in CMJ ranges between 25 and 27%. However, the results mainly depend on the sport and the level of the subject: PE-students, beginners or advanced, high-performance sports athletes [45]. In Laffaye’s [45] research, skilled men jumpers performed about 39 cm higher than females. 

The results also showed that men achieved the same height of 29 cm in both jumps (from 30/60 cm box), but women obtained only 2 cm higher jump from a 40 cm drop-off (25 ± 0.5 cm) compared to the 20 cm box. Additionally, men achieved a greater jump height by moving their COM higher than women during the ground contact phase of the jump. The jump height was probably due the same time of take-off, which differed ~0.01–002 s in favor of women (Table 2 and Table 3). According to Kirby et al. [46] and McMahon et al. [39], it is possible to attain this by the development of greater velocity of take-off (concentric phase) by men, even with their larger contact with the ground. It appeared that men in jumps from both heights (30/60 cm) achieved very similar contact times with the ground and reached the same value of take-off velocity. In turn, women reached different COM displacement in both jumps, which also means a different rebound-jump height (difference was 5 cm). The jump height was affected by the same support phase time (Table 2 and Table 3), but probably a different take-off velocity, which is dependent on the concentric net impulse and given mass of the athlete [39]. The body mass differed significantly between men and women, p = 0.000 and d = 2.34. 

In addition to the height of the jump, three kinematic parameters, relative maximal F, contact time, left leg—relative maximal F, right leg—relative maximal F, showed a significant (p > 0.05) difference in the values between men and women, in both jumps, for each group. The finding showed that the relative maximal force reached in both the eccentric and concentric phases of the drop-CMJ jump significantly discriminated between men and women, both in the jump from the lower box 20/40 cm and higher box 30/60 cm. In jumps from the higher box, men exceeded women by 17.5% in terms of force. It is also interesting that the differences in the relative maximal force were at the same level between 15 and 17% in jumps within the group, in favor of the jump from the higher box. These findings were contrary to research done by Riggs and Sheppard [47], Rice et al. [31], and McMahon et al. [39], who found no discrimination between men and women in force production; however, they referred to the measurement of relative peak force. The reason for the different forces attained by the sexes is probably due to the different jump strategies performed by each group. However, this strategy mainly depends on box height, and mainly, differences in body mass. Undoubtedly, the height of the box can be compared to the ground contact time in each jump with more or less effect. However, we must remember that the contact time in both groups differed in favor of women, i.e., was shorter. This probably affected the landing and the size of the leg stiffness and its ability to absorb the force [48]. However, this ability was not measured in this experiment. This statement agrees with McMahon et al. [39], who claimed that women and men adopt differential leg stiffness strategies during jumping tasks where higher leg stiffness was reached by women.

An interesting trend was shown by the measurement of relative maximal F divided into left and right limbs compared to double legs, but only in women. The left limb showed a higher maximum force compared to the right limb in both jumps (DJ20/DJ40 cm box). The men showed the same trend only when jumping from a lower drop-off height. These differences probably resulted when landing with a faster contact with the ground (platform) was shown by the left limb, which may be associated with the fact that most right-handed athletes performed take-off from the left limb, which seems more effective [7]. The deficit of force production during take-off between a single leg and bilateral measurement was at the level of 1.5% among both women and men in three jumps—from 20 cm and 30/60 cm, respectively. Women with a jump height of 40 cm showed no deficit. This proved our hypothesis, i.e., that homogenous groups, either female or male, might demonstrate similar side-to-side (right-to-left) differences in force production compared to double-leg rebound jumps. 

In contrast to the relationship of force production, this experiment did not find significant differences between sexes in the relative power output for jumps of lower (20/30 cm) and higher (30/60 cm) drop-off heights. This is a quite a surprise because previous studies have shown that the increase in power and differences in value occurred in men. For example, Laffaye and Choukou [38] found higher values of the relative mean power for males (56.9 6 25 W/kg) against 42.4 6 19 W/kg for females, representing a difference of 25.5%. This usually happens because males have a higher velocity during take-off (concentric phase of the jump), which causes an increase in relative power. In our experiment, we can only assume values of the relative mean power, because we did not measure the velocity of a drop-countermovement jump but relied on the statements of current literature. This statement confirms that the difference in jump performance between sexes [48] could be explained by the difference of the force parameters [34,38], and vertical jump height depends on the vertical velocity at the take-off, which is correlated with the power output [33,39].

Based on this analysis, it can be understood that the drop-off height application in the drop-rebound type of jump depends on the task itself. It is supposed that a DJ with lower 20/30 cm or higher 40/60 cm (women/men) height can emphasize either the force or power output via an increase in the velocity component in the rebound action. The force increases when the ground contact time lengthens, which depends not only on the drop-off height but mainly on the angle in the knee joint when landing after the drop. Therefore, force increase can happen when the duration of the ground contact is not a limiting factor in jump performance. In turn, when the ground contact time is the main focus, which means an increase in power output in rebound-jumps, the velocity component in the concentric phase is a primary factor. For example, Walsh et al. (2004) reported that increasing the drop height from 0.2 m to 0.6 m led to a change in GRFmax by the same value as jumps from 0.2 m, but performed very quickly (contact time = 0.14 s) and performed much more slowly (contact time = 0.21 s).

It also should be assumed that the velocity component may increase in value along with the greater differences in athlete’s muscle mass (differences in sex). The muscle mass overloads the neuromuscular capacity of athletes in a manner that facilitates more elastic energy in the muscle–tendon system. More elastic energy causes a quicker release during the push-off phase to accelerate the males’ body and reach a higher velocity during the entire phase of take-off than women; even the ground contact time differs significantly between men and women. This occurred in this experiment. The effect is transferred to the jump performance and does not necessarily have to influence the height of the jump. However, this is contradicted by previous research of Komi [15], who claimed that women showed a better ability to use a larger percentage of the energy stored during the pre-stretching phase of the drop-rebound jump.

Another view of the drop-rebound jump is that the increase in velocity at take-off, i.e., shortening of the contact time, determines the intensity of DJ by the intensification of an eccentric load. According to Potach and Chu [49] and Pedley et al. [24], this intensification is directly influenced by the duration of the exposure to gravitational acceleration (height of drop-off). This is confirmed by Flanagan and Comyns [41] and Wilson and Flanagan [50], following Peddley et al. [24], who in their research stated that the increase in box height might impact velocity, which may subsequently generate greater power and loading rates. This can happen if the jump performance task exceeds the athlete’s eccentric force-producing capacities [51]. It is known that an increase in drop-off height, followed by its impact on jump execution, may expose an athlete to injury [51,52,53]. Therefore to avoid such an eventuality, it is desirable to use a “one control“ drop-off height, which allows minimizing the risk of injury, but maximizes the overall jump performance, either by maximizing the height of the jump or by maximizing the task imposed during a DJ jump.

Despite the results presented herein, we are aware the present study has several limitations. Our results could be different if the population would be comprised of elite athletes. Athletes were from a variety of sports, and we could not control in which part of the season are they as some of them were in the preparatory period vs. some of them being in the competition period. Another limitation may be the use of a different box height, which did not allow for the complete comparison of the obtained kinematic parameters between women and men. However, the main goal of the study was to find one box height and obtain from it: either by maximizing the height of the jump or maximizing the task imposed during a DJ jump.

## 5. Conclusions

High relative maximum force and power production in a DJ is possible from both a lower drop-off height; 20 cm for women and 30 cm for men, and from a higher one—40 cm and 60 cm, respectively. This means that the drop height of 20–60 cm was shown to be a quantifiable factor to reach the value of such kinematics parameters that are necessary to overcome the CMJ resistance in order to improve jump performance (height). The mentioned drop height will also minimize the risk of injury, but maximize the overall jump performance, either by maximizing the height of the jump or by maximizing the task imposed during a DJ jump. In both jumps, the left limb showed greater maximum force compared to the right limb. Women showed significantly greater differences than men. The higher the jump height box (DJ20/DJ40 cm), the greater the difference in strength. These results did not fully confirm our hypothesis. This study also indicates that coaches should take care to control their athletes’ drop jump technique when prescribing drop jumps as the maximum load of plyometric workouts. The increase of training load should not happen by a direct increase (even gradual) of drop-off height but by application of optimal—“one control“ drop-off height, contained between 20 and 60 cm.

## Figures and Tables

**Table 1 ijerph-17-05886-t001:** Descriptive statistics of selected anthropometric characteristics of female and male athletes.

Variables		*N*	Mean	Std. Deviation	95% Confidence Interval				
					Lower Bound	Upper Bound	f	*p*	*d*
Age [year]	Men	44	20.29	1.32	20.20	21.64	3.541	0.0039	1.55
	Women	20	21.18	1.29	20.50	21.91			
Body height [cm]	Men	44	182.19	6.43	180.23	184.14	87.573	0.000	2.56
	Women	20	166.78	5.29	164.30	169.26			
Body mass (kg)	Men	39	78.65	7.09	76.35	80.95	69.974	0.000	2.34
	Women	20	62.23	7.21	58.86	65.61			
SMM (kg)	Men	39	40.76	4.14	39.41	42.10	167.836	0.000	3.44
	Women	20	27.51	2.67	26.26	28.76			
Percent body fat (%)	Men	39	9.72	2.61	8.87	10.56	136.932	0.000	3.27
	Women	20	20.34	4.37	18.29	22.38			
Body mass index	Men	39	23.36	1.59	22.84	23.88	4.227	0.044	0.47
	Women	20	22.28	2.39	21.17	23.40			

**Table 2 ijerph-17-05886-t002:** Descriptive statistics of selected kinetic variables of CMJ during drop jumps from 20 cm (female) and 40 cm (male) athletes.

Kinetic Variables		*N*	Mean	Std. Deviation	95% Confidence		F	*p*	*d*
					Lower Bound	Upper Bound			
Drop Height [cm] Drop Height [cm]	Men	44	30				Between Groups		
	Women	20	20						
Jump height from Flight T [m]	Men	44	0.29	0.06	0.22	0.31	17.312	0.000	1.06
	Women	20	0.23	0.05	0.21	0.25			
Relative maximal F [%BW]	Men	44	663.75	103.01	632.43	695.07	7.428	0.008	0.75
	Women	20	593.05	78.61	556.26	629.40			
Relative maximal P [W/kg]	Men	44	78.09	32.51	68.21	87.98	0.683	0.412	0.22
	Women	20	70.97	30.62	56.65	85.30			
Relative E [J/kg]	Men	44	4.78	2.88	3.91	5.66	0.203	0.654	0.09
	Women	20	4.45	2.47	3.29	5.60			
Contact T [s]	Men	44	0.19	0.03	0.18	0.20	0.993	0.323	0.37
	Women	20	0.18	0.02	0.18	0.94			
Relative maximal F [%BW] – left leg	Men	44	333.91	58.04	316.26	351.56	6.915	0.011	0.72
	Women	20	296.25	39.72	277.66	314.84			
Relative maximal F [%BW] – right leg	Men	44	337.50	54.57	320.91	354.09	4.575	0.036	0.58
	Women	20	305.45	57.74	278.43	332.47			

**Table 3 ijerph-17-05886-t003:** Descriptive statistics of selected kinetic variables of CMJ during drop jumps from 30 cm (female) and 60 cm (male) athletes.

Kinetic Variables		*N*	Mean	Std. Deviation	95% Confidence		F	*p*	*d*
					Lower Bound	Upper Bound			
Drop Height [cm] Drop Height [cm]	Men	44	60				Between groups		
	Women	20	40						
Jump height from Flight T [m]	Men	44	0.29	0.06	0.27	0.31	7.252	0.009	0.74
	Women	20	0.25	0.04	0.23	0.27			
Relative maximal F [%BW]	Men	44	801.20	140.06	758.62	843.79	7.854	0.007	0.77
	Women	20	699.35	121.97	642.29	756.41			
Relative maximal P [W/kg]	Men	44	82.19	38.21	70.57	93.81	1.769	0.188	0.36
	Women	20	69.36	29.55	55.53	83.19			
Relative E [J/kg]	Men	44	6.05	4.77	4.59	7.49	2.224	0.141	0.41
	Women	20	4.37	2.29	3.29	5.44			
Contact T [s]	Men	44	0.21	0.03	0.19	0.22	3.435	0.069	0.68
	Women	20	0.19	0.03	0.18	0.21			
Relative maximal F [%BW]- left leg	Men	44	406.64	69.87	385.39	427.88	10.029	0.002	0.87
	Women	20	346.95	69.92	314.23	379.67			
Relative maximal F [%BW] – right leg	Men	44	406.82	84.74	381.05	432.58	5.393	0.024	0.71
	Women	20	357.60	62.46	328.37	386.83			

**Table 4 ijerph-17-05886-t004:** Comparison of the Spearman rank correlation coefficients calculated for selected somatic parameters and kinematic parameters of RJ at p < 05.000.

DJ 20 cm								Women	DJ 40 cm							
**(8)**	**(7)**	**(6)**	**(5)**	**(4)**	**(3)**	**(2)**	**(1)**		**(1)**	**[2]**	**[3]**	**[4]**	**[5]**	**[6]**	**[7]**	**[8]**
−0.13	**−0.54***	**0.52***	**−0.47***	**−0.49***	−0.38	−0.08	0.12	Body height	0.12	−0.02	−0.13	−0.16	−0.17	0.07	−0.18	−0.04
−0.09	**−0.69****	**0.63****	**−0.49***	**−0.58****	**−0.44***	−0.16	−0.20	Body mass	−0.17	−0.29	−0.41	−0.35	−0.27	0.39	**−0.45***	−0.33
DJ 30 CM								Men	DJ 60 cm							
**(8)**	**(7)**	**(6)**	**(5)**	**(4)**	**(3)**	**(2)**	**(1)**		**(1)**	**(2)**	**(3)**	**(4)**	**(5)**	**(6)**	**(7)**	**(8)**
−0.04	−0.04	0.19	0.12	0.04	−0.06	0.16	0.13	Body height	0.15	−0.08	0.01	0.04	0.01	−0.10	0.03	0.15
−0.07	−0.04	0.24	0.14	0.06	−0.07	0.07	−0.04	Body mass	0.03	−0.27	−0.03	0.03	0.18	−0.27	−0.21	0.03

(**1**) Drop Height, (**2**) Jump height from flight, (**3**) Relative maximal F, (**4**) Relative maximal P, (**5**) Relative E, (**6**) Contact T, (**7**) Left leg—Relative maximal F, (**8**) Right leg—Relative maximal F. * *p* = 0.05, ** *p* = 0.01.

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
