# Peer review of "The Effect of Height on Drop Jumps in Relation to Somatic Parameters and Landing Kinetics"

_ijerph, 2020, doi:10.3390/ijerph17165886_

Round 1

Reviewer 1 Report

The study design was straight-forward for comparison of some mechanic parameters between genders. The description of the protocol and methods is clear. If the purpose of the study was to show the gender differences between jumping height and muscle force generation, it’s well served.

My question came from the underlying association between those somatic parameters and risks for muscle injuring when performing plyometric activities. The discussion part is mostly descriptive to further accentuate the gender difference without giving enough rational on whether to minimize the difference or to identify the risks of muscle injuries among genders.  The discussion focused on minimizing risk of muscle injury came later on with limited support of research evidence and did not clarify if the incidence of injury or any risks is associated with gender differences in jumping height and landing kinetics.  My suggestion is to keep your discussion cohesive and elaborate the gender-associated musculoskeletal injuries with existing research evidence with linkage to the current research data.

Please also kindly clarify:

Table 1-3. The column of “No”, number seems be confusing to me. Base on the description (Page 3, Line 3), there were “forty male” participants. The table shows 44 for males.   

Reviewer 2 Report

                                                                      2nd of August 2020

Journal: IJERPH (ISSN 1660-4601)

Manuscript ID: ijerph-892126 (Article)

Title: The effect of drop jump height and selected somatic parameters on landing kinetics

Authors:  Krzysztof Mackala, Samo Reuter , Janez Vodociar , Jozef Simenko , Robi Kreft , Jacek Stodolka , Jozef Krizaj , Milan Coh

REVIEW REPORT

This article analyzes and compares between both sexes, the effect that fall height has on performing Drop Jumps (DJ). Through a statistical correlation analysis of some somatic parameters with selected kinetic variables of the classic countermovement jumps (CMJ), on the landing kinetics in these jumps, it is shown that the fall height of 20-60 cm is a factor to take into account to achieve improvements in kinematic parameters, which are necessary to overcome resistance in CMJs, in order to improve jump performance.

Data collection was carried out through the execution of 3 jumps for each participant, using two independent and synchronized force platforms (S2P Bilateral Tensiometric Platform) in each of the 2 assigned jump heights (20 and 40 cm for women and 30 and 60 cm for men).

The article is correctly structured in each of its parts, although it would be enriching to know the applicability of the study and its limitations and to translate them into the article.

Regarding the title, in the statement proposed by the authors, the importance of height is clear, but it does not correctly express the difference between somatic parameters (height, body mass SMM,%, BMI) and kinetics (Flight, F max, P max ...). For the title to be adequate it would have to have a grammatical structure of this type: “The effect of height on the Drop Jump in relation to somatic parameters and landing kinetics”.

The introduction provides sufficient background to contextualize the article and the references are relevant and up-to-date. At the end of the introduction, reference is made to the hypothesis of the work (page 2 Line 48) that would be more appropriate to put before stating the objectives of the study made earlier in the text (page 2 Line 45).

Regarding the research method, the study population that appears in section 2.1 Participants, quantifies 20 female physical education students and 40 male, this last data does not coincide with the 44 male athletes that appear in data collection from tables 1, 2 and 3.

The research design and measurement procedure is adequate and well explained in the text, perfectly understanding the development of the process, carried out in accordance with the code of ethics of the World Medical Association and with the corresponding permits from the Human Ethics of the University of Ljubljana (Code: 14_2019-1436).

In the results section, the descriptive statistics of the four tables clarifying the somatic factors and the selected CMJ kinetic variables, which are announced in the objectives, clarifying the aspects of force and power production.

Regarding the discussion, at the beginning of the discussion there is an error in one of the figures that establishes the height of the high drawers at 30 and 60, being the entire article at 40 and 60 (Pag.7 Line 2). References from other authors are adequate and in line with the investigation. Some reference misquoted (Pag.7 Line 47).

To finish the structural part, the conclusions do not refer to the hypothesis from which the objectives are born, in which special emphasis is made on the difference in the lateral leg in the production of force and power, displacement time ..., aspects that they are not specified in the conclusions. If aspects related to sports injuries and recommendations to coaches are considered in this section, these are not conclusions that are directly derived from the study and that they could be applications of the study or other lines of research.

STRENGTHS OF THE ARTICLE

  • New vision of work comparing the heights of the box and sex of the athletes that contributes concrete results to apply in the redeeming sport.
  • Good structuring of the article.
  • Current references on the same subject
  • Clarity in the results

WEAKNESSES OF THE ARTICLE

  • The participants are non-athlete students, therefore it will have to be taken into account when extrapolating the results to the elite sport.

  • The sports of origin of the participants can condition the results obtained, due to the physical and technical requirements of each sports specialty.
  • Decompensation in the number of participants by sex
  • The objectives are defined in different sections of the article (Summary, introduction and discussion) in a similar way.
  • The conclusions do not meet one of the objectives of the study.

In conclusion, the article is good for publication, but I recommend improving the errors detected and reflected in blue in the text of this report and taking into account the suggested improvement contributions.
